# The Internet, the Problem of Socialising Young People, and the Role of Religious Education †

David Kraner

**Abstract:** Alongside the declining religiosity of young Slovenians, there is a growing loneliness among young people. When young people are not motivated or do not have the opportunity to engage in social activities in their free time, they look elsewhere for substitutes. In our study, we highlight the problems young people face with their loneliness, their excessive use of the internet, their low involvement in social activities, and their high tolerance for smartphone distraction. Religious education in Catholic grammar schools in Slovenia plays an important role not only in providing religious content, but also in empowering adolescents to take a critical view of the world, and to actively engage young people in society.

**Keywords:** internet; loneliness; socialisation; leisure; religious education; young people

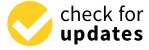



## 1. Introduction

The internet is neither good nor bad in itself, yet it is not neutral. Young people, who are one of the most vulnerable groups in society, need to be prepared to face the aggressive intrusions of the internet. In this respect, the school, and through it, religious instruction, has an important educational role to play.

School has always been an educational institution, and should continue to play this role. In the past, it was enough to involve young people sufficiently in leisure activities (sports, social engagement, volunteering, etc.) to promote their socialisation. Today, however, other ways must be found to bring young people into contact with each other.

Society today offers many opportunities for young people, but it does not provide them with educational guidance and important normative control. Problems arise when young people, who do not yet have a strong identity, find themselves in a complex social environment without orientation. The family and the school are marginalised by social media and the power of corporations (Vodičar 2021, p. 895). The family no longer has a monopoly on education (Platovnjak 2020, pp. 364–69).

The space, time, and place in which we live, allow many agencies to enter this field. That is why education today is plural. Young people today spend their free time outside the family. In some cases, the family is almost completely absent. Educators and parents must be aware that they are not the only actors in education. Today, social networks have a strong influence on education (Bajzek 2008, pp. 29–31).

Many schools, as well as various government policies in many countries, have programmes to educate young people in media literacy. However, Buckingham notes that many schools have fallen into 3 traps: 1. A defensive or protectionist approach (debates about media violence, the impact of media on addiction, obesity and consumerism), 2. Political anti-propaganda (demystification of the media, replacing 'fake' messages with 'real' ones), and 3. Creativity (spreading the myth of creativity, that young people—the digital natives—are different from others) (Buckingham 2020, p. 84).

Media literacy is not only about knowing how to use certain technological devices to access and/or create media content. It must also necessarily include an in-depth critical understanding of how media work (function), how they communicate, how they represent (represent) the world, how they are produced, and how they are used. To understand media today is to recognise the complexity of contemporary forms of 'digital capitalism'. And if we really want to have competent citizens, we need comprehensive media education programmes that are systematically supported as a basic right for all boys and girls (Buckingham 2020, p. 26).

A critical view of media content must be underpinned by hope and expose the aforementioned enemies of hope: cynicism, fatalism, relativism, and fundamentalism. All of this is possible if we have the understanding that the media are not transparent, that they are not a window on the world that reflects reality, but that they are representations of the world. Buckingham presents four areas of analysis: media language, representations, production, and audience (Buckingham 2020, pp. 68–70).

Slovenia excludes denominational religious instruction in public schools, unlike Austria, Croatia, Italy, Germany, Poland, and elsewhere in Europe. It is officially classified as a country where non-denominational religious instruction is practised. A small number of pupils take such classes (religion and ethics), however, "de facto in Slovenia, it is difficult to speak of the presence of religious instruction in public schools" (Globokar 2019, p. 124).

Poland is the opposite of Slovenia in terms of religious education for young people. Poland is one of the most Catholic countries in Europe: 92% of children in primary schools receive religious instruction. In secondary schools, however, only 70% of young people choose to take religious instruction. Makosa et al. find that the main reason for avoiding religious instruction is opposition to or disagreement with Catholic teaching on sex education (Mąkosa et al. 2022, p. 8).

In Slovenia, the situation is quite different. Surveys of young people (aged 16–27) show that the proportion of religious people has fallen by 30% in the last 20 years (2000 to 2020). Only 44% of young Slovenians consider themselves to be Catholic. The number of young people who do not belong to any religious community has increased by 20% over the same period (Lavrič et al. 2021, p. 383).

In Slovenian Catholic grammar schools, where religious instruction is provided, a very wide range of topics is covered. In the first grade, religious instruction focuses on interpersonal relationships, culture and religion. In the second grade, the Bible, the meaning of life and the personal journey of salvation are discussed. In the third year, the different religions and sects are discussed. In the fourth year, the role of the media in society, the importance of the family, and current issues in bioethics are examined.

Confronting such a breadth of life issues allows young people to look at their faith and religious affiliation from different perspectives, while at the same time providing topics of interest to non-believers or non-Catholics. Religious education is also about moral development, which is intrinsically linked to cognitive, emotional, and social development. "The objective of moral development is an independent and responsible person who recognises the other as a person with the same rights and duties, who is capable of reasonable decisions and is able to choose the good for himself/herself, for society, and for the entire natural environment. A moral personality makes decisions in an autonomous, reasonable, and responsible way" (Globokar 2018, pp. 553–54).

A large survey of parents in the UK in 2016 found that social networks can spread both negative and positive attitudes. Negative traits include the spread of hate, arrogance, ignorance, and judgement. Positive ones include humour, admiration for beauty, creativity, love, courage, and kindness (Jubile Centre 2016).

Communication is fundamental to Christianity and other religions. God communicates with man, and man with his fellow man (Platovnjak and Svetelj 2022, pp. 630–35). We are interested in the world of young people and their communication through the internet in order to better understand their behaviour. To this end, we interviewed young people in Slovenian Catholic high schools about their use of social networks and chat services, their

feelings of loneliness, their motivation for leisure activities, and the whereabouts of their smartphones during their studies.

We hypothesised two things: 1. Young people in Catholic high schools are exposed to the internet for more than an hour a day and have a high tolerance for smartphone distractions while studying. 2. Young people in Catholic high schools do not feel lonely and are very actively involved in leisure activities.

The survey was conducted from 7 to 25 November 2022 in 4 Catholic grammar schools in Slovenia, covering 672 young people aged between 18 and 19 years.[1]

## 2. Young People's Exposure to Social Networks and Chat Services

The internet has become part of who we are and how we communicate, so it is important to find the best ways to integrate it into young people's lives in a useful way. On the one hand, we have research showing the negative impact of the internet on our lives. This shows that overexposure to social networks has the most negative impact on life satisfaction in young people up to 19 years of age (Orben et al. 2022, pp. 1–10). Girls are at higher risk of mental health problems (depression, self-harm, eating disorders) than are boys (Salk et al. 2017, pp. 783–822).

On the other hand, we have experts who emphasise the positive value of media, if we integrate it properly into our lives. Ten years ago, Henry Jenkins argued that young people have great potential to develop their own digital competencies, that they belong to a culture of participation and are highly creative in the digital sphere (Jenkins 2014, pp. 60–61).

Rather than looking at positive and negative impacts, we need to look at the context in which we live. Jenkins (2014) and David Buckingham found that the biggest problems were for those who work in schools, who were much less media literate compared to the younger generations they work with (Buckingham 2010, p. 55). Buckingham criticises the polarised debates that, on the one hand, see technology as the solution to all problems (cyber-utopianism, drilling and skilling, empowerment), and on the other hand those that blame the media for all problems in society (cyber-bullying, sexting, fake news, trolling, flaming, filter bubbles). The problems are much broader, and have a history in the antecedent media (comics, television, cinema, the polemical press) (Buckingham 2020, pp. 50–55).

According to a 2016 survey in Slovenia, young people spend 2.3 h a day on a computer or tablet, and 3.4 h on a mobile phone. It is telling that most young people do not resort to delinquent behaviour. The most common forms of unwanted behaviour among young people are cheating on tests and conflicts with parents (Rek 2021, pp. 32, 76).

In our survey, we wanted to know how much time young people spend on the internet each day. We distinguished between internet use on a PC and on a smartphone. Young people spend much more time using the internet on their smartphones. In the 'Very often', category, that is, 3 to 6 h a day, 35% of young people are exposed to the internet on their smartphones, compared to only 7% on their PCs. For 'Very rare', that is, less than 1 h a day, 12% of young people are exposed to the internet on their smartphones, compared to 56% on their PCs. Finally, in the 'Frequently' category, or 1–3 h a day, 46% of young people are exposed to the internet on their smartphones, while 33% are exposed to the internet on their PCs (Figure 1).

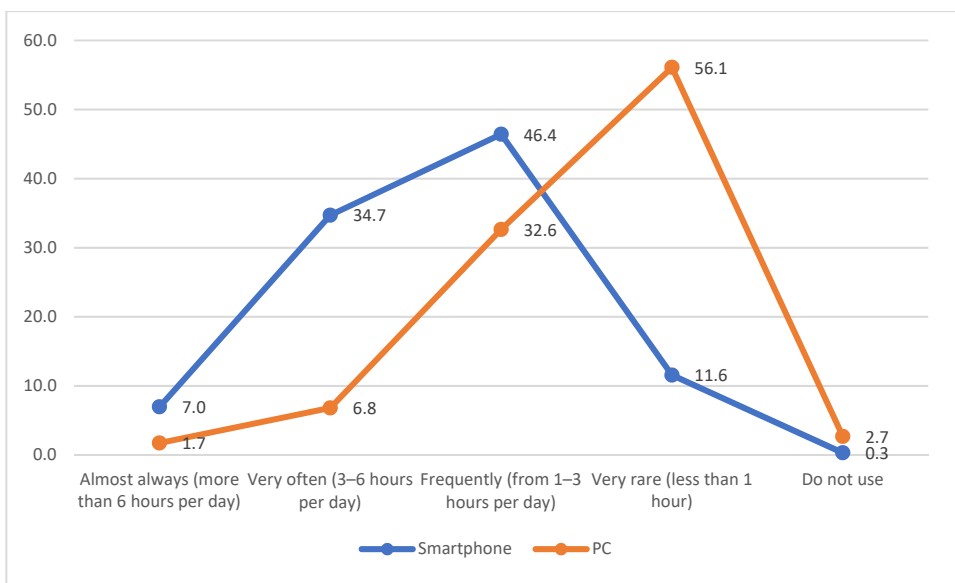

**Figure 1.** Exposure to internet.

Based on these data, we can assume that young people mainly use smartphones for chatting, while PCs are more likely to be used as a learning tool.

As young people are more exposed to the internet via smartphones than PCs, we investigated whether there are any gender differences. The data show that 3% of girls are more exposed to the internet via smartphones. However, there are no statistically representative differences by gender in terms of exposure to the internet via PC. We found that 87% of boys and 89% of girls use the internet daily via the smartphone. Also, 6% of boys and 8% of girls use it for more than 6 h a day. We also found that 36% of boys and 34% of girls use the internet very often (3–6 h), while 46% of boys and 47% of girls use the internet frequently (1–3 h per day), and 12% of boys and 11% of girls use the internet rarely (less than 1 h) (Figure 2).

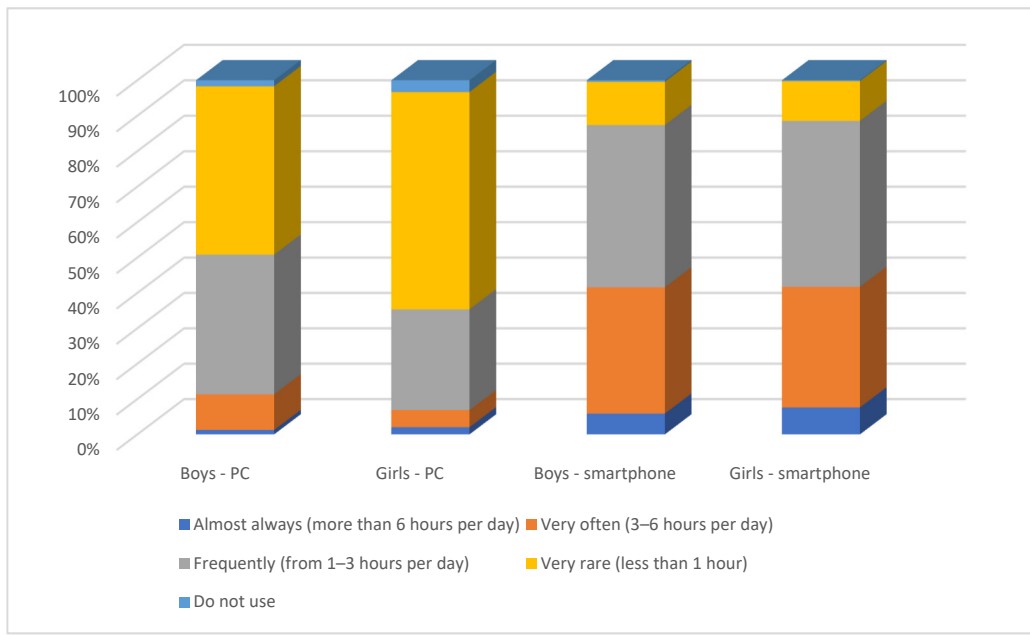

**Figure 2.** Exposure to internet via smartphone and PC by gender.

We were also interested in what young people do on the internet or how much time they spend using different social networks. Instagram is the most used (90%), followed by

Facebook (55%), Pinterest (46%) and TikTok (44%). For Facebook, Instagram and Pinterest, most of the young people who use them do so very rarely, less than 1 h per day (Instagram 36%, Facebook 30%, Pinterest 35%, and TikTok 11%). Only Instagram has a high number of users (27%) in the 'rarely' category, 1–2 h per day, and LinkedIn, Tinder and Tumblr are the least used (Figure 3).

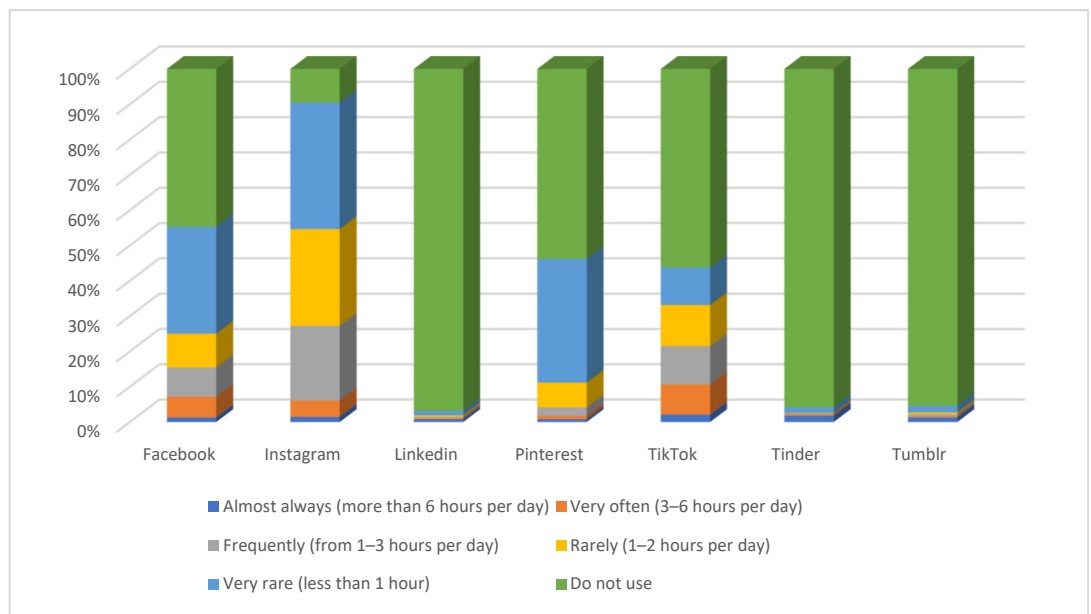

**Figure 3.** Exposure to social networks.

Among chat services, Snapchat (72%), Viber (52%), and WhatsApp (33%) are the most used. As with social networks, the largest proportion of young people who use chat services do so for less than 1 h a day ('very rare') With the exception of Snapchat, 29% of young people use it for less than 1 h, 17% 'frequently' (up to 3 h per day), and 8% 'very often' (for 3–6 h a day). Viber is also used less than 1 h daily. Similar proportions are also found for WhatsApp, which is used for less than 1 h a day by 27% of young people. We also found that 29% of young people use a smartphone for less than 1 h a day, 17% used one often, and 8% used a smartphone for 3–6 h a day (Figure 4).

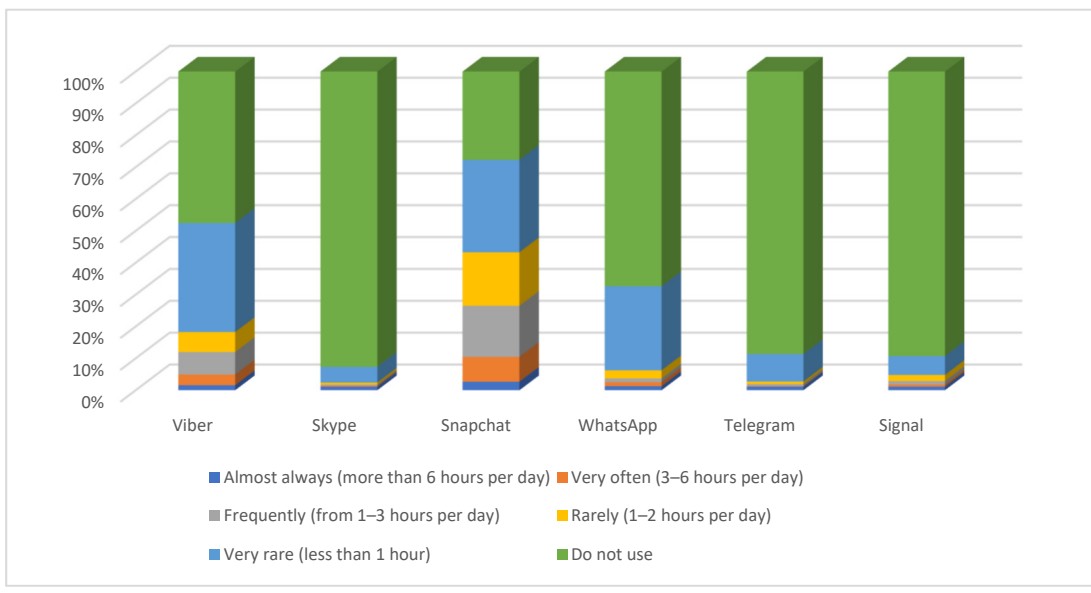

**Figure 4.** Exposure to chat services.

The context in which young people live is crucial in interpreting the results. "When the Internet is used as a way station on the route to enhancing existing relationships and forging new social connections, it is a useful tool for reducing loneliness. But when social technologies are used to escape the social world and withdraw from the "social pain" of interaction, feelings of loneliness are increased" (Nowland et al. 2017, p. 70).

In their study, Primack and colleagues found that social media has a greater negative impact on those who use it for more than two hours a day, than on those who use it for less than half an hour a day (Primack et al. 2017).

To summarise the use of the internet, the following can be highlighted: most young people use the internet on a daily basis via their smartphones, and spend most of their time on Instagram, Facebook, Pinterest, and TikTok, among social networks, and Snapchat, Viber, and Whats App, among chat services. Of these social networks, most users use them for less than an hour a day. There is a notable percentage of young people who are overexposed to the internet (6% of boys and 8% of girls). The lowest percentage of young people exposed to the internet (half an hour per day) is 12%.

## 3. Loneliness

Hikikomori is a syndrome that refers to a phenomenon whereby people choose to live completely isolated from the real world (Japanese: to stand alone, to isolate oneself). Japanese psychiatrist Tamaki Saito uses this word to describe a phenomenon in which people isolate themselves from social life, due to the excessive use of digital media (Amendola et al. 2018, pp. 52–53).

Certain social media events can foster feelings of exclusion, and idealised representations of peers' lives can also foster envy and the misconception that others are happier (Primack et al. 2017, pp. 6–7). Real face-to-face relationships are more challenging than online relationships. According to Globokar, on the internet, "relationships are less risky, easier to withdraw from, less painful; in real life, building genuine interpersonal relationships requires hard and demanding work, but these are the only fully human relationships. In real life, interpersonal verbal communication is always accompanied by a context of gestures, facial expressions, tone of voice and other forms of non-verbal communication, all of which are required to create a real relationship" (Globokar 2019, pp. 76–77).

Withdrawal from society and exclusion from active social life are typical of depressed people. When people spend time together, talking, exchanging opinions, sharing experiences, thoughts, and feelings, they engage with each other directly. We feel the emotions of our interlocutor from their voice, facial expressions, smell, etc. We cannot experience this in front of screens. Just as we can only learn to walk or talk from direct contact with other people, so we can learn empathy only from direct contact. Over-exposure to the internet reduces empathy in adolescents (Spitzer [2018] 2021, pp. 28–29).

Recent research confirms that excessive internet use contributes to loneliness (Reed et al. 2023; Orsolini et al. 2023; Islam et al. 2023; Shi and Wang 2022; Spitzer [2012] 2017, p. 24).

Our study does not show the impact of the internet on loneliness. Nevertheless, we are interested in the general feeling young people have about their level of loneliness: 2% of young people said they are always completely lonely, 23% of young people are often completely lonely, 61% of young people are rarely completely lonely, and 14% are never completely lonely (Figure 5).

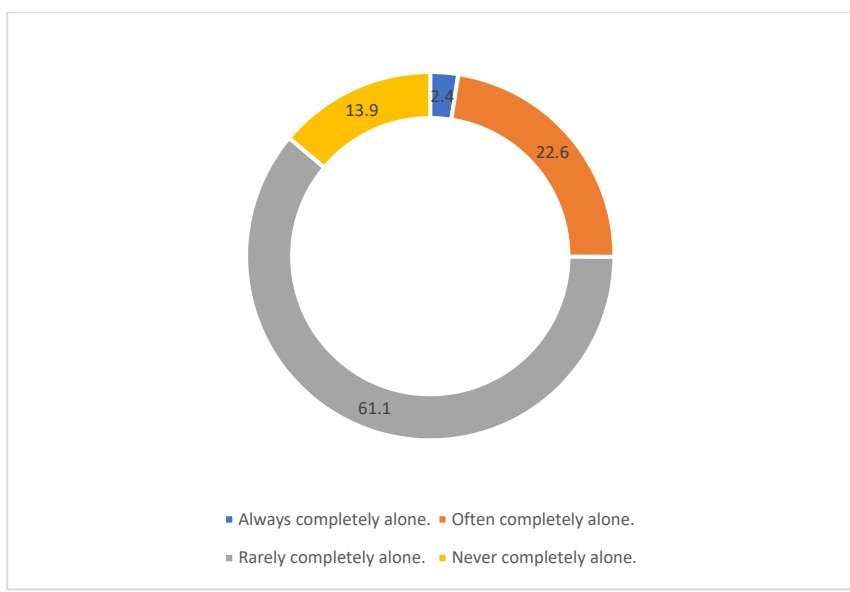

**Figure 5.** Loneliness.

In terms of gender, girls stand out. Among them, 27% are always and often completely lonely (22% for boys), while only 12% of girls (and 18% of boys) are never lonely (Figure 6).

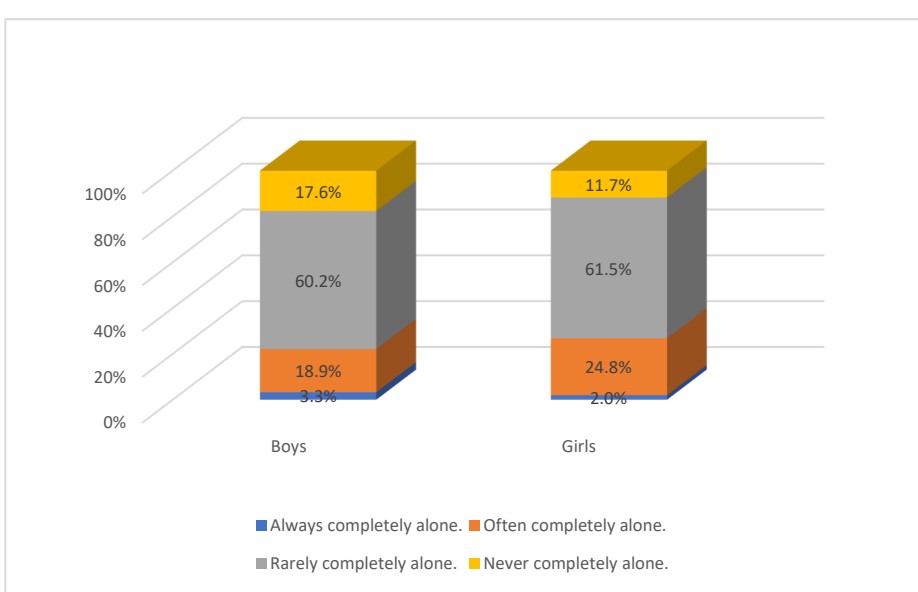

**Figure 6.** Loneliness by gender.

In our analyses, we find that most young people feel "rarely completely alone", regardless of how much time they spend using the internet. We only see differences between those who are the most exposed to the internet, and those who spend less than an hour with the internet. Among those exposed to the internet the most, 11% of students answered that they feel always completely lonely, while among those less exposed to the Internet, 25% answered that they feel never completely lonely. Among those who are 'often' and 'very often' exposed to the internet, the proportion of those who answered that they feel lonely very often, stands out (Figure 7).

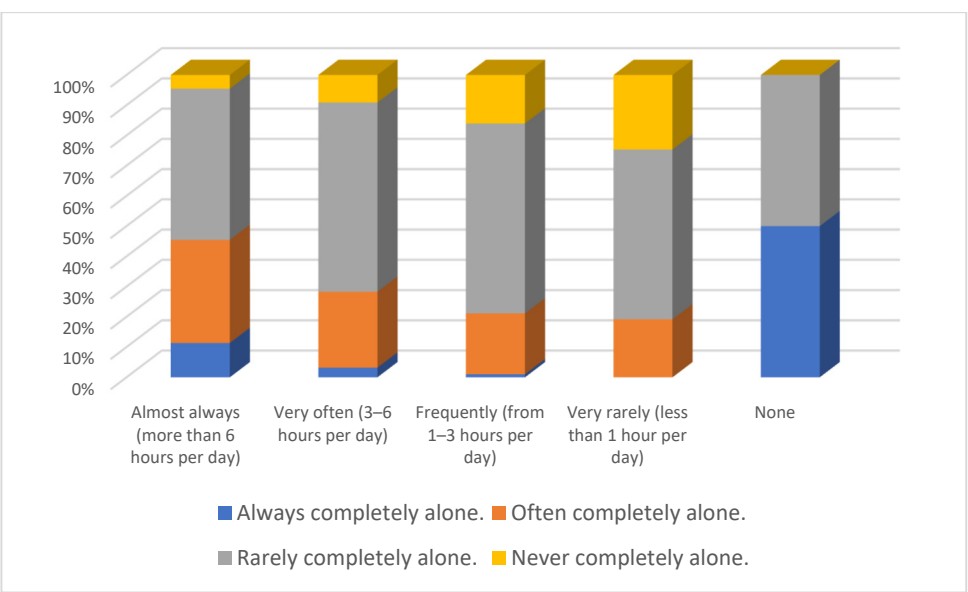

**Figure 7.** Proportion of lonely students by exposure to the internet on the phone.

In summary, most young people "admit" that they "rarely feel completely alone". The higher number of lonely people is found among girls. No significant correspondence was found between the duration of internet use and the feeling of loneliness.

## 4. Motivating Young People to Engage in Leisure Activities and Socialisation

Digital media makes it possible to communicate, connect, and stay connected. We can agree with Globokar that internet networking that does not build real relationships is of little value (Globokar 2019, p. 76). Therefore, encouraging students to engage in various activities in their free time, is one of the very important tasks of today's educators and parents.

We mentioned hikikomori syndrome. This disorder can develop in three stages. In the first stage, the student seeks reasons to avoid live contact with people (absent from school, dropping out of activities, disrupted sleep-wake rhythms). In the second stage, he also refuses invitations to socialise with friends, and spends most of the time in his room (losing sleep-wake rhythms). In the third stage, the student finds himself in complete isolation (at high risk of psychopathologies) (Amendola et al. 2018, p. 53).

Social networks are designed to connect, communicate, and share. Yet, despite all the connectivity, students feel isolated. Researcher Nowland argues that when the internet is used as a space to deepen existing relationships or make new social connections, it is a useful tool to reduce loneliness. However, when the internet is used to escape from the social world, feelings of loneliness increase (Nowland et al. 2017, p. 70).

Social activities, face-to-face relationships, and in-person friendships, offer more satisfaction to young people than social networks. Over-exposure to the internet, takes time away from personal hobbies (reading books, sports, theatre, etc.), necessary housework (helping to clean rooms, washing dishes, gardening, etc.), etc. Research confirms that smartphones reduce time for sporting activities (Kim et al. 2015; Lamberg and Muratori 2012).

Our survey assessed how much time young people spend on personal hobbies, leisure commitments, and volunteering activities. Personal hobbies stand out, with 90% of young people spending an hour or more on them each day. This is followed by leisure time commitments at home, and then weekly voluntary activities (Figure 8).

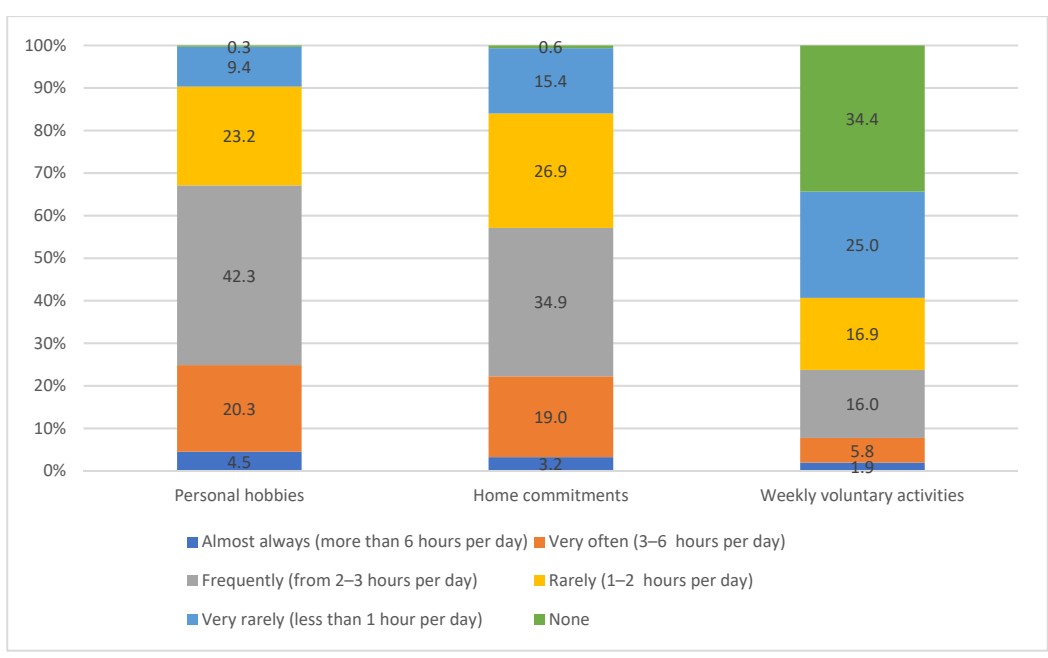

**Figure 8.** Student leisure activities.

Other research shows that young people's interest in culture is declining (Mesojedec et al. 2014; Črnič 2018; Grilc et al. 2016, p. 55). A survey of young people aged 15 to 20, shows that young people's interest in cultural activities declined on average between 2000 and 2020. The ratings, on a 6-point scale (1—never; 6—every day), are as follows: decreasing attendance at cultural institutions—cinema, theatre, concerts (2000—2.9; 2010—2.5; 2020—2.3); very poor attendance at museums and theatres (2020—2.1); lower participation of young people in cultural creation (2000—2.9; 2010—2.4; 2020—2.7); and lower activity in writing—diary, poems, letters, interest in reading (2000—2.3; 2010—1.5; 2020—1.8) (Lahe and Cupar 2021, p. 247).

Tanja Oblak Črnič, professor of communication studies, notes that one third of young people are very socially active, and two thirds are passive. Passive young people do not practice anything in the field of cultural and media consumption. These are young people who have lower school performance and parents with secondary education. Socially and culturally active young people consume events and content while also actively participating. In addition, they have experience of traditional cultural institutions. These young people achieve the highest grades in school and have educated parents (Črnič 2018, pp. 53–54).

In our research, however, we see that young people are active in those activities where they have the greatest extrinsic and intrinsic motivation. Personal hobbies are somewhere in the middle between young people's personal interest and external obligation (expectations and wishes of parents or teachers). Compulsory activities are set by the student's parents or educators. This is an extrinsic motivation. Involvement in voluntary activities, on the other hand, requires the greatest intrinsic motivation (someone is a member of a football club because they enjoy football). In terms of intrinsic and extrinsic motivation, we see that young people are most involved where both are present, and least involved where it is purely about their intrinsic motivation.

In our survey, 54% of young people do not volunteer even 1 h a week. However, 88% of young people spend an hour or more on the internet on their phone, and 41% on the internet on their computer, on a daily basis.

What is the reason for such low engagement of young people in voluntary activities? We find (in our survey) that young people are active where they are "forced to do something". This means that, in addition to personal motivation and talents, there needs to be an external incentive. Hull, who saw the link between behaviour, drive, instinct, and incentive, already spoke of incentive. He drew up the following equation about the role of

incentives in behaviour: behaviour = drive × habit ×/+ incentive (Hull 1951). The two signs ("×" and "+") are in the equation because it is not quite certain whether the influence of incentives is more often additive, or multiplicative.

The incentive value of a goal, gratification, or reward, has a strong motivational power, similar to that of drive and habit (Grum et al. 2009, p. 138). Therefore, teachers and educators should have a very good knowledge of young people on the one hand, and on the other hand, of the techniques of encouraging young people. Students would become more socially engaged. However, we should not forget that there are many distractors.

## 5. The Internet as a Distraction from Work

Our concentration on work (learning) is affected by a variety of distractions: unpleasant ambient noise, people in the room, personal biological needs, and even the sound or light of smartphones and computers. Unlike all the others, we can turn off or remove these devices from our field of vision or space at any time.

Research shows that the physical proximity of smartphones affects our concentration. The mere presence of a smartphone reduces or limits cognitive abilities, even when the user manages to concentrate on a task (Roberto et al. 2015; Booker et al. 2015, pp. 173–79).

The new media operate without pause, constantly churning out a stream of information that does not allow our minds to stop and take stock. This flow is followed by mental capacities that are impaired and in a state of dependence and inertia (Larchet [2016] 2022, p. 188). Digital devices lead to superficial thinking and to stress. They cause unwanted side-effects, ranging from pure distraction, to child pornography and violence (Spitzer [2012] 2017, p. 87). The proliferation of information in short, disconnected, often overlapping bursts (the faster the better) has shaped a new way of thinking. This is the opposite of focused, continuous, linear thinking (Carr 2011, p. 119). People who frequently use multiple media at the same time show problems in controlling their minds. Multitaskers perform significantly worse than non-multitaskers. Even when it comes to task switching, which is common among multitaskers, multitaskers are significantly slower than non-multitaskers" (Spitzer [2012] 2017, p. 112).

Focus is the attachment and concentration of consciousness in a limited time on one of the mental operations: perceiving, thinking, remembering, reflecting, etc. Focus is the opposite of distraction and dispersion, change, disequilibrium, momentariness, and the simultaneous performance of several tasks at the same time (Larchet [2016] 2022, p. 194). The focus of the mind is on one of the mental operations: perceiving, thinking, remembering, reflecting, etc.

In our study, we wanted to know whether young people carry their smartphones in their line of sight, out of their line of sight, or not at all, while they are learning. The results show that only 14% of young people carry a smartphone outside the place of study (always or often), 52% of young people carry it out of sight, and 67% of young people carry it in sight (Figure 9).

The sound of the phone is also a distraction. We found that 66% of young people always or often have the sound on when studying, 80% when expecting a call or message, and 62% when not expecting a call or message (Figure 10). At work, 83% of girls and 74% of boys always or often have the sound on.

This data shows that young people are very attached to their smartphones. It is a strong desire to be always reachable and to have all the information from their "friends". However, this does not always have positive effects. Young people who are always connected to the internet (homo connecticus) find it difficult to dispose of time frames. This is already confirmed by the data on young people's inactivity in their free time and the constant presence of the phone. Larchet is convinced that the media environment encourages constant distraction: "The time available to them is no longer marked by duration in the form of a temporal sequence, but by a series of confused moments" (Larchet [2016] 2022, pp. 195–96).

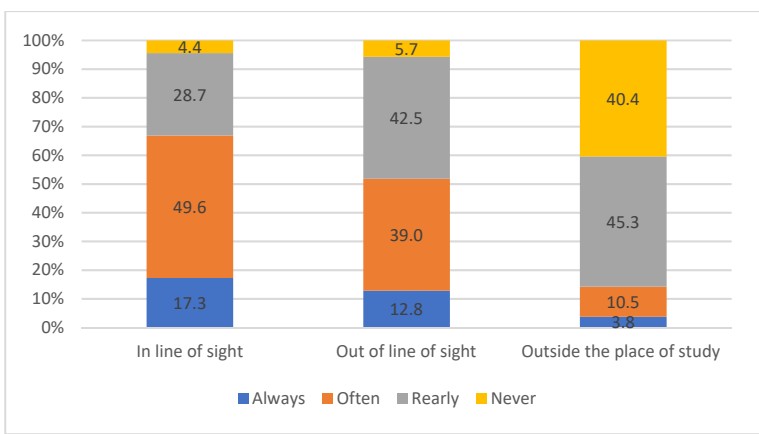

**Figure 9.** Phone range at work.

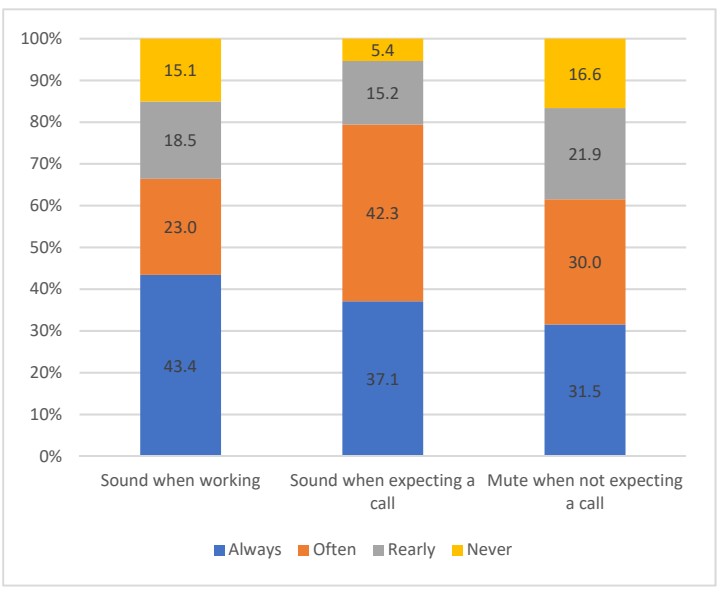

**Figure 10.** Phone sound at work.

The fast pace of life and overwork are therefore not only experienced by adults, but also by young people. Borut Škodlar, a professor at the Ljubljana Psychiatric Clinic, believes that calming exercises should also be practised in schools. "A child who is distracted, unfocused, needs to calm down first" (Bojc 2023).

If exposure to the internet (playing online games) is not regulated in time, addiction can result. Various studies clearly show the negative effects of internet addiction on physical and mental health. Internet addiction is associated with levels of anxiety, depression, and aggressive behavior in young people (Choi et al. 2018, p. 549).

Here, the role of religious teachers and educators is very challenging and important. Young people are not aware of the powerful influence the internet has on them.

## 6. Self-Perception of the Impact of the Internet on Interpersonal Relationships

Students are convinced that the internet does more good to them than harm, when it comes to socializing. They believe it does more harm to others than to themselves. Only 28% of young people 'admitted' that they personally think the internet is harmful (somewhat harmful, very harmful, or quite harmful) when it comes to socializing with their peers. However, when asked if the internet is harmful to others when it comes to making friends, 38% of young people answered "yes" (Figure 11).

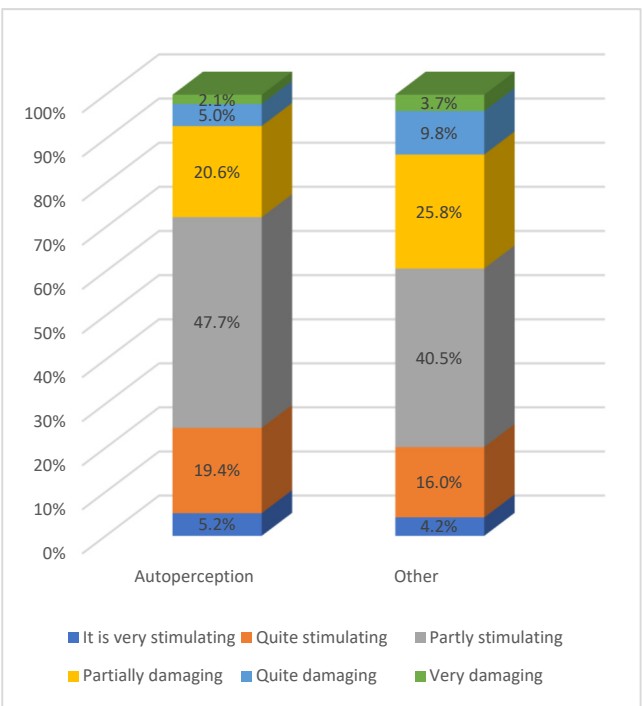

**Figure 11.** Self-perception versus perception of others.

Differences in self-perception are also evident between boys and girls: 31% of boys and 20% of girls answered that the internet is beneficial in making friends ('very' and 'quite'); 6% of boys and 8% of girls admitted that the internet is ('very' and 'quite') harmful in making friends with peers. Boys find the internet more stimulating, while girls find it more harmful (Figure 12).

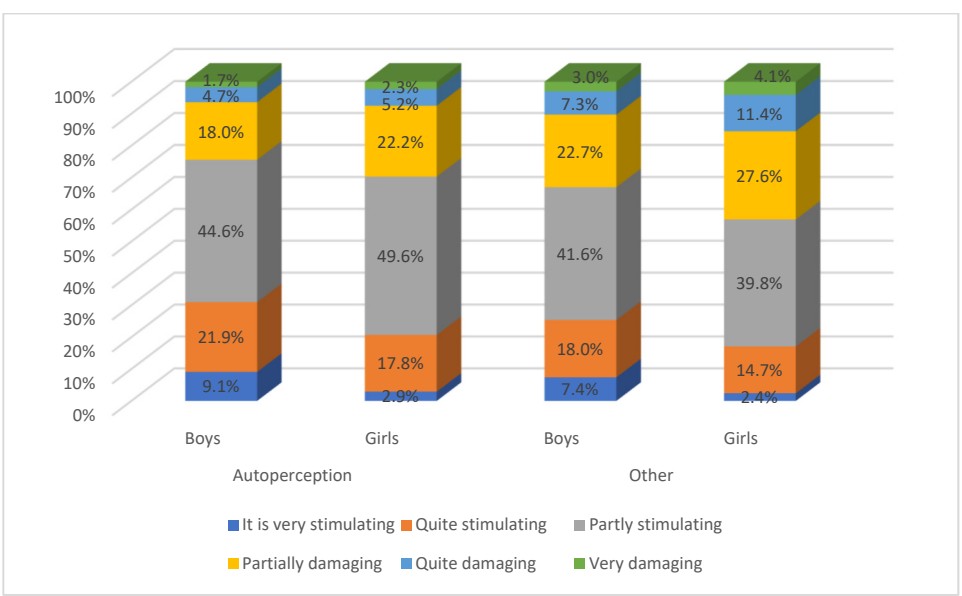

**Figure 12.** Perception of the impact of the internet by gender.

## 7. Conclusions

The results of our investigation confirm the first hypothesis and reject the second. The first hypothesis is confirmed: young people are highly exposed to social networking and chat services (more than one hour per day), and young people are very tolerant of light and noise distractions from their phones while studying. In fact, we find that young people are

connected to the internet most of the time through their phones, which are always within reach, even when they are studying. Almost all of them are connected to the internet for an hour every day. Most young people use Instagram and Snapchat.

The second hypothesis, which assumed that young people in Catholic grammar schools do not feel lonely and are very actively involved in leisure activities, was rejected. The results of our survey show that very few (less than half of the young people) volunteer for 1 h per week (fire brigade, parish youth work, Caritas, etc.), although 84% of young people do at least 1 h of voluntary housework per day. Almost all young people pursue private hobbies for at least one hour per day. We found that 23% of young people often feel lonely (2% feel completely lonely). One third of young people are aware of the harmful effects of the internet on socialising, and more than one third have difficulties socialising.

Not only young people, but also teachers and educators, are under the strong influence of distractions. They are so powerful that they can get in the way of our achieving the desirable goals of education. That is why teachers and educators should first work on themselves, in order to be able to pass on good practices to young people (Jeglič 2022, p. 734; Nežič Glavica 2019). Just as we have found that young people need encouragement, so too do teachers and educators. The European projects should devote more attention and material support to this area.

Confessional religious instruction is a great challenge today. Good catechesis necessarily involves personal experience. "Only a catechesis that proceeds from religious information to personal guidance and to the experience of God will be capable of offering meaning. The transmission of the faith is based on authentic experiences, which must not be confused with experiments: experience transforms life and provides keys for its interpretation, while the experiment is reproduced only in an identical manner" (Pontifical Council for the Promotion of the New Evangelization 2020, p. 131).

Losé Luis Moral, a pastoral theologian at the Salesian Pontifical University, argues that young people have difficulty understanding what we call faith, and that they do not reject faith, but our ideas about faith (Moral 2007, p. 120). Moral stresses that three aspects are important in the evangelisation of young people: "(1) The logical and profound intertwining of the Christian experience with the experience of young people in the context of the 'communication age'. (2) Taking into account the decisive importance and weight of culture in religious education. (3) Identifying the main reason for the progressive alienation of young people from the Church" (Moral 2007, p. 121).

The task of the modern religious teacher is also one of continuous formation and personal spiritual growth. In addition to a good knowledge of the content, the teacher must know his students and the environment in which they live (including social networks, chat services and other websites that young people visit). We cannot ignore the fact that young people are lonely, have difficulties in participating in leisure activities, and difficulties in socialising. Religious education, with its many topics, can help address these problems among young people.

In the future, it will be necessary to carry out a complete overhaul of both religious education textbooks and additional training for religious education teachers. In the digital age, not only the way we communicate has changed, but also the human being himself. The old concepts for communicating knowledge and faith are no longer sufficient to answer the modern young person's questions about meaning, and we do not yet have new ones.

**Funding:** This research was funded by *Javna agencija za raziskovalno dejavnost Republike Slovenije (ARRS)* (en. Public Agency for Research of the Republic of Slovenia) grant number J6-4626 (B). And The APC was funded by Republic of Slovenia.

**Institutional Review Board Statement:** The study was not submitted to the Ethics Committee for the following reasons: The questionnaire does not contain any depth or psychological questions. The topics are exposure to the Internet, use of social networks during learning and leisure time, social engagement, and perception of the impact of the Internet.

**Informed Consent Statement:** After a personal consultation, one member of the Ethics Committee told me that this type of question does not need the Ethics Committee review. Voluntary participation in the survey was ensured. The survey was conducted in complete anonymity. All data is protected.

**Conflicts of Interest:** The author declares no conflict of interest.

## Note

[1]   A survey was carried out as part of this project. The questionnaire does not contain any depth or psychological questions. The topics are: exposure to the internet, use of social networks during learning and leisure time, social engagement, and perception of the impact of the internet. Voluntary participation in the survey was ensured. The survey was conducted in complete anonymity. All data is protected.

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
