# Peer review of "The Internet, the Problem of Socialising Young People, and the Role of Religious Education"

_religions, doi:10.3390/rel14040523_

Round 1

Reviewer 1 Report

I am attaching my comments in the pdf below. 

Overall, the article lacked sufficient focus in terms of one of its stated aims to address what RE can contribute to the empowerment of adolescents.

The language in the article was overly subjective at times.

There was insufficient evidence of the metrics used to make certain judgements in the article e.g. that adolescents are not over-exposed to the internet. 

Claims were made throughout the article which were not backed up with evidence. 

The graphs which were presented were difficult to read and one was labelled incorrectly. 

There were several typos. 

There were contradictions inherent in some of the statements. 

Author Response

Dear Sir or Madam.

Thank you for your valuable comment. I am sending a corrected and updated text.

Author

Reviewer 2 Report

The title of the article is too general and does not precisely refer to the content of the article, which claims to be based on a survey carried out in Slovenia. The title lacks such information. The author has not formulated the research objective. The reader does not know what the author wanted to achieve, prove or demonstrate. The social survey conducted among young people in Catholic schools in Slovenia has nothing to do with the conclusions formulated. In any case, no such survey is necessary for such conclusions. They are self-evident and have already been established many times, as the author himself proves with abundant references. As a result, the article contains many general statements that sound trivial and are certainly not the conclusions of the surveys conducted.

Author Response

Dear Sir or Madam.
Thank you for your comments.
I am sending a corrected and amended text.

Author

Reviewer 3 Report

I recommend this fine article with two concerns.  

First, the research design (survey) and data collection are sparsely described. As you see, I raised a question of ethics. We absolutely need to see how the young people were invited, chosen and protected. Who administered the questionnaires and how was it collected. Ethical protections and human subject review and description must be included in at least a substantial footnote if not in the text.

Second, I’d love to see more suggestions of next steps. Now that this excellent information has been garnered, what specific suggestions could be made to educators to address how we now know youth access and use social media.

It is an outstanding article and well written.  I appreciate the way they weave scholarly publications and insights with their research. The information is important!  How education and religious education can assess and respond to the worlds of young people is crucial. This study could help us do that. That is why I suggested some expanded attention to engagement with finds for teachers. 

Well done!

Author Response

(The authors gave the same response as above.)

Round 2

Reviewer 1 Report

Some of the comments which were submitted after the first review have been attended to. 

On the whole though, the presentation of the data, the manner of commentary is still not satisfactory.  Tables are poorly presented and some are labelled incorrectly. One is the wrong graph entirely (graph 5). 

In addition, claims are made in the research around causation with no evidence presented to support the claims e.g. the research found that certain percentages of those who engaged in internet activity were lonely, but the claim can't be made that their engagement in the internet activity was what was making them lonely or what caused their loneliness. In addition, claims are made concerning addiction also which imply a causative factor between internet usage and addiction. This claim is disputed in recent literature on addiction.  

The potential contribution of Religious Education receives little attention even though it is suggested in the abstract that this is a focus of the article. 

Author Response

Dear Sir or Madam.
I have thoroughly reviewed your comments and evaluations and have corrected the text. "Please see the attachment."
Kind regards.

Reviewer 2 Report

The corrections and adaptations made have improved the quality of the text, which is now more coherent and meets the stated objective. I now have no disqualifying objections.

Author Response

Thank you for your comment, as I was able to improve the point I wanted to make in the article.

Round 3

Reviewer 1 Report

I have included my comments on the attached file.

I think that the article has been improved since the previous version. The inclusion of hypotheses provides a much better cohesion to the article. However there are still issues with regards to the presentation of data and discussion of findings. In addition, the stance of the writer/s is not clearly enunciated. 

Author Response

ok
